# Workplace Violence among Healthcare Providers during the COVID-19 Health Emergency: A Cross-Sectional Study

**DOI:** 10.3390/bs12040106

**Published:** 2022-04-13

**Authors:** Othman A. Alfuqaha, Nour M. Albawati, Sakher S. Alhiary, Fadwa N. Alhalaiqa, Moh’d Fayeq F. Haha, Suzan S. Musa, Ohood Shunnar, Yazan AL Thaher

**Affiliations:** 1Jordan University Hospital, The University of Jordan, Amman 11942, Jordan; 2Department of Nursing, Jordan University Hospital, The University of Jordan, Amman 11942, Jordan; noorbwati@gmail.com (N.M.A.); sakher_horizon@yahoo.com (S.S.A.); ohood.shunnar@gmail.com (O.S.); 3Faculty of Nursing, Philadelphia University, Amman 19392, Jordan; fhalaiqa@philadelphia.edu.jo; 4School of Medicine, Jordan University Hospital, The University of Jordan, Amman 11942, Jordan; mohammedhaha1@gmail.com (M.F.F.H.); suzan.ramini@gmail.com (S.S.M.); 5Faculty of Pharmacy, Philadelphia University, Amman 19392, Jordan; yalthaher@philadelphia.edu.jo

**Keywords:** communication skills, COVID-19, workplace violence, nurses, physicians

## Abstract

(1) Background: Workplace violence among healthcare providers (HCPs) is a tangible barrier to patient care. The purpose of this study was to gain insight into physicians’ and nurses’ perceptions of workplace violence and their perceptions of communication skills during the COVID-19 health emergency. We also sought to assess and compare the association between types of workplace violence, communication skills, and several sociodemographic factors of physicians and nurses including gender, marital status, site of work, age, and educational level during this era. (2) Methods: We performed a cross-sectional study of a selected hospital in Jordan using the Arabic version of the workplace violence and communication skills scales for September to November 2020. We included a total of 102 physicians and 190 nurses via a self-reported questionnaire. (3) Results: During the COVID-19 health emergency, physicians (48%) experienced workplace violence more than nurses (31.6%). More than two-thirds of the participants did not formally report any type of violence. Multinomial logistic regression analysis showed that marital status, gender, age, site of work, educational levels, and communication skills were associated with different types of violence among the two samples. (4) Conclusions: A high prevalence of workplace violence is noted among HCPs in Jordan compared with before the pandemic, which highlights the importance of promoting public awareness during crises.

## 1. Introduction

Workplace violence among healthcare providers (HCPs) is a tangible barrier to patient care. This barrier may negatively affect hospitals and create additional burden risks to patients [1]. However, prior to the coronavirus disease 2019 (COVID-19) pandemic, public sector physicians in both Jordan and China faced different types of workplace violence. [2,3]. The current research results on the prevalence of this phenomenon in the university hospital sector are poorly investigated during this era.

Since there are known associations between workplace violence and sleep quality [4], anxiety [5], and the incidence of post-traumatic stress disorders [6], the possible COVID-19 disease burdens could increase workplace violence among physicians and nurses. Moreover, the increase in the COVID-19 burden could decrease their professional status [7], increase their unwillingness to serve patients, as well as their fear of dealing with patients [8]. A growing body of evidence highlights the problem of workplace violence against physicians and nurses.

Physical abuse, verbal abuse, and nonverbal abuse are all identified as the main types of workplace violence. Compared with other types of violence, more than 50% of HCPs have experienced verbal abuse [5,9]. Little attention has been paid to different types of violence during the COVID-19 health emergency. Sociodemographic factors including gender, marital status, site of work, and age were found to be associated with workplace violence among HCPs. For example, it was found that nurses suffer from workplace violence approximately two times more than physicians [10]. Generally, male gender and lower educational levels among HCPs were more likely to be related to the experience of workplace violence compared with female gender and higher educational levels [5,11]. A similar finding was reported in India, where the youngest physicians had the highest levels of workplace violence [12]. The perceived level of workplace violence among HCPs in emergency rooms and psychiatric wards was higher than in other healthcare sectors [13]. This study will better represent the relevant sociodemographic factors of physicians and nurses by exploring their contribution to workplace violence during the era of COVID-19.

We hypothesized that a higher perceived level of communication skills among HCPs can provide higher productivity and success, lower anxiety levels, higher patient satisfaction, better decision-making, and lower workplace violence [14]. In contrast, we hypothesized that lower communication skills among HCPs may anticipate higher physical violence, lower motivation, and lower commitment, and may affect the quality of care and services provided to patients [15,16]. Studies focusing on the association of communication skills and workplace violence among physicians and nurses during this era are limited.

Given the debate over the effect of workplace violence and communication skills on physicians and nurses, it is unclear which occupations will experience more workplace violence. Recruited samples were selected based on their strenuous job with COVID-19 patients and based on their extensive contact with patients and patients’ families during the COVID-19 health emergency. This study will highlight the occurrence of workplace violence among medical professionals (physicians vs. nurses) during the COVID-19 health emergency. However, this study aimed to gain insight into physicians’ and nurses’ perceptions of workplace violence and their perceptions of communication skills. Moreover, we hypothesized the relationship between types of workplace violence and communication skills. Finally, we sought to assess and compare the association between types of workplace violence, communication skills, and several sociodemographic factors of physicians and nurses including gender, marital status, site of work, age, and educational level during this era. This study provides important information for identifying the occurrence of workplace violence among HCPs during the era of the COVID-19 pandemic.

### Study Questions

What are the occurrences of workplace violence and the perceived level of communication skills among physicians and nurses during the COVID-19 health emergency?Is there an association between the types of workplace violence and communication skills among physicians and nurses during the COVID-19 health emergency?What are the sociodemographic factors affecting workplace violence among physicians and nurses during the COVID-19 health emergency?

## 2. Materials and Methods

### 2.1. Study Design

We used a cross-sectional design during the period of the COVID-19 pandemic among physicians and nurses in a selected hospital in Jordan. We ensured adherence to the STROBE checklist guidelines.

### 2.2. Participants and Setting

All physicians and nurses in the selected hospital in Amman-Jordan were invited to participate in our self-reported survey. The selected hospital has a sufficient number of physicians and nurses, and it is considered a center for COVID-19 patients. Exclusion criteria were physicians and nurses who had maternity leave, did not hold a degree, were on unpaid leave, or had long periods of sick leave. The total number of physicians in the selected hospital was approximately 350, while the number of nurses was approximately 850 [2,17]. The sample size was calculated based on the precision of 0.04 with a confidence interval of 95%. In this regard, a total of 114 participants would be representative of each profession [18]. Based on the number of physicians to nurses, we distributed 150 and 250 self-reported surveys, respectively. A convenience sampling procedure was selected to recruit the study sample. We went to each supervisors’ site of work and asked them to distribute it among their staff. Participating sites of work were the emergency room, adult floors (medical/surgical), the pediatric floor, intensive care units, and the psychiatric unit. The number of returned surveys for physicians totaled 115 with a response rate of 76.7%. A total of 13 surveys were excluded due to incomplete surveys, extreme values, or contradictory answers; thus, the analysis of physicians’ participants was performed on 102 participants. Regarding nurse participants, a total of 190 nurses completed valid surveys with a response rate of 76%.

### 2.3. Data Collection

We distributed self-reported surveys from September to November 2020. Voluntary participation, a consent form, data protection, and anonymous identification were ensured. Before beginning the study, ethical approval was gained by the institutional review board in the selected hospital.

### 2.4. Study Tools

#### 2.4.1. Sociodemographic Factors

This section refers to the personal information of physicians and nurses including gender, marital status, site of work, age, and educational level. Various sites of work were chosen and are presented in Table 1.

#### 2.4.2. Workplace Violence Scale

This scale was used to measure the prevalence of workplace violence among HCPs, which was adopted from Bayram et al. [19]. It consists of two parts: The first part measures the experience, frequency, and type of violence (5 items). The second part assesses security precautions against violence (6 items). Table 2 reveals the detailed items of the workplace violence scale. We translated the selected scale from English to Arabic using forward translation, backward translation, and bilingual expert opinion. This process was determined by 8 experts (PhD holders) in nursing, medicine, the English language, and psychology. They completed the final Arabic version and agreed on 100% of all items; hence, none of the items were omitted. The validity process was determined by factor loadings > 0.40 [20], Kaiser–Meyer–Olkin (KMO) Test > 0.60, and Bartlett’s test of sphericity by calculating the Chi-square value (χ^2^) and *p*-value < 0.05 [21]. In the analyses, factor loadings ranged between 0.46 and 0.87, the KMO test was 0.66, and Bartlett’s test of sphericity was (χ^2^: 384.511; df: 55; *p* < 0.001). The total Cronbach alpha was 0.80. Hence, the Arabic version of the workplace violence scale is valid and reliable. We rated this scale based on a Likert scale (e.g., no, once, more than once; yes/no).

#### 2.4.3. Communication Skills Scale

This scale was used to measure the perceived level of communication skills (12 items), which was adopted from Axboe et al. [22]. The translation process was completed by the selected experts. Factor loadings ranged between 0.59 and 0.95, the KMO test was 0.94, and Bartlett’s test of sphericity was (χ^2^: 4322.473; df: 66; *p* < 0.001). The total Cronbach’s alpha was 0.97. We rated this scale based on a 5-point Likert scale. For positive items, the scale was: 5 “Always”, 4 “Often”, 3 “Sometimes”, 2 “Rarely”, 1 “Never”. Negative items were rated in the reverse order.

### 2.5. Statistical Analysis

The data were analyzed by the statistical package for social science (SPSS V. 22). Therein, the normal distribution was checked by the chi-square test. The variables related to workplace violence and communication skills, means, standard deviations, and frequencies were determined. To assess the differences between both groups, the independent-sample t-test was used. Multinomial logistic regression analysis was applied to achieve the association of sociodemographic factors with types of workplace violence (physical, verbal, and nonverbal abuse). All *p*-values of 0.05 were significant.

### 2.6. Ethical Consideration

We first obtained ethical consideration from the institutional review board at the selected Hospital (No. 10/2020/19073). Given the concern for ethical principles, the Declaration of Helsinki guidelines were followed.

## 3. Results

### 3.1. Demographic Characteristics

A total of 190 nurses and 102 physicians completed the self-reported surveys during the era of COVID-19. Table 1 shows the sample characteristics and chi-square test between the two professions based on their sociodemographic factors, including gender, marital status, site of work, age, and educational level.

According to the chi-square test, both groups’ data were normally distributed since the chi-square values were significant for all selected sociodemographic factors. By comparing the socio-demographic information in the two groups, nurses were generally female and married, aged between 31 and 40 years old, and more than two-thirds of them had bachelor’s degrees. In terms of the site of work, 58 nurses were working on the medical/surgical floor, 51 were in the emergency room, and the remaining nurses were working in different units (Table 1). Conversely, physicians were generally single, male, aged between 20 and 30 years old, and 61 had higher educational levels. Approximately half of the physicians were working on medical/surgical floors. A total of 16.7%, 15.7%, and 10.8% of physicians were working in the emergency room, the pediatric floor, and the psychiatric unit, respectively.

### 3.2. Occurrence of Workplace Violence and Perceived Level of Communication Skills

To assess the occurrence of workplace violence (physical, verbal, and nonverbal abuse), security precautions, and the perceived level of communication skills, frequencies, percentages, means, standard deviations, t-test, and statistical significance levels are illustrated in Table 2.

Table 2 indicates that physicians and nurses experienced more than one type of violence during the era of COVID-19. Physicians (48%) experienced workplace violence more than nurses (31.6%). Among the three types of violence, verbal abuse was the most common, with a percentage of 87.3% for physicians and 69.5% for nurses. Physical abuse was the lowest percentage in both professions. There were significant differences between physicians and nurses in terms of verbal and physical abuse. Results indicated that nurses are often physically abused, while physicians are often verbally abused. Generally, physicians and nurses did not formally report any type of workplace violence. Metal detectors, police checkpoints, and security guards were found to be at lower percentages. Overall, there was no presence of security precautions against workplace violence except security cameras. Regarding the perceived level of communication skills, high levels of communication skills were found in physicians, with 61.8%, compared to moderate levels in nurses, with 41.1%, and this difference was statistically significant (Table 2).

### 3.3. Association Factors with Type of Workplace Violence

Table 3 presents the association of workplace violence including physical, verbal, and nonverbal abuse with sociodemographic and communication skills. Multinomial logistic regression analysis was applied accordingly.

#### 3.3.1. Physicians

Our results indicated that marital status, site of work, age, and educational level were positively associated with physical abuse. The model also showed that site of work, age, and communication skills were positively associated with verbal abuse. Finally, age and communication skills were positively associated with nonverbal abuse among participating physicians.

#### 3.3.2. Nurses

We found that the site of work for nurses was positively associated with physical abuse. Moreover, site of work, age, and communication skills were positively associated with verbal abuse. Gender and communication skills were positively associated with nonverbal abuse among participating nurses.

## 4. Discussion

With the continuous evaluation of the COVID-19 health emergency, the burden of COVID-19 is ascendant. In this study, we have assessed the occurrence of workplace violence among physicians and nurses. We found that almost half of the participating physicians and one-third of the participating nurses had been subjected to workplace violence more than once. Physicians are more likely to experience workplace violence than nurses. Verbal abuse exhibits the highest prevalence in both professions. A lack of security precautions is found, except security cameras were reported to be present. Communication skills are found to be at high levels among physicians and moderate levels among nurses. Among both groups, gender, site of work, age, and communication skills are all associated with different types of violence.

### 4.1. Occurrence of Workplace Violence and Perceived Level of Communication Skills

HCPs have experienced more than one type of workplace violence during the COVID-19 health emergency. This dilemma becomes challenging to HCPs and directly affects their regular duties and tasks, thereby affecting their patients’ lives. This study agrees with several previous studies [1,23]. Physicians are significantly suffering abuse more than nurses and it may occur due to patients’ and families of patients’ beliefs that decision-making, the information given, and problem-solving are mainly related to physicians. In the same vein, the long period of stay among COVID-19 patients, inadequate information, and impatience are the main reasons behind workplace violence against HCPs in Turkey [24]. During this era, nurses have less contact with patients than physicians who are directly involved with patients in emergency rooms, intensive care units, and medical/surgical wards. Our findings can help to explain the evidence reported in other studies of higher turnover and lower productivity among Jordan physicians [2]. This study is congruent with a study conducted in Saudi Arabia (KSA), which found that workplace violence against physicians (47%) is higher than against nurses (41%) [25]. Comparing our results with the non-pandemic period, previous studies in Jordan [26,27] showed that nurses experienced verbal violence (63.9%) compared to those in our study (69.5%). This study highlights the increase in verbal abuse during the COVID-19 health emergency. In 2019, a review of 253 studies found that verbal violence was the most common, with a percentage of 57.6% [28]. Our finding is in agreement with a previous longitudinal study in Italy [29], which found that physicians suffer a higher occurrence of workplace violence than nurses. Moreover, it found that the rate of physical violence among physicians was (19%) compared to those in our study (19.6%) [29]. Longitudinal studies reflect real-world situations of workplace violence in HCPs as well as during the pandemic period.

Notably, more than two-thirds of participants in this study did not formally report any type of violence. This could be related to the lack of security precautions, lack of system, the nature of the disaster, lack of information on how to officially report, and a sense of uselessness when they report violence against them. This finding is consistent with a study conducted in KSA [30]. Indeed, the lack of security precautions has been strongly linked to frequent violence among healthcare providers [19]. The presence of security precautions provides a sense of control, a safe place, and a secure environment for them. Appropriate policies and interventions from the director of the hospital to tackle the security precautions are strongly needed.

Interestingly, the perceived level of communication skills among physicians is higher than in nurses during the COVID-19 health emergency. Potential explanations are linked to the fact that these skills are self-reported and that this might lead to a somewhat contradictory finding. Essentially, good communication skills render a significant effect on patient satisfaction [31], mitigate disruptive physician behavior [32], enhance self-efficacy, and reduce burnout levels [33,34].

### 4.2. Association Factors with Type of Workplace Violence

We found that verbal abuse was the most common type of abuse among both samples in this study. This pattern of violence was previously reported among HCPs in Switzerland [35]. Among physicians, different sociodemographic factors were positively associated with physical abuse including marital status, age, educational level, and site of work. They experience this type of violence more frequently than nurses because of their workload, time pressure, critical condition of COVID-19 patients, and the increased number of deaths. Meanwhile, in nurses, only the site of work factor was associated with physical abuse, and this can be related to workplace factors such as working in the emergency room or psychiatric unit.

This study also showed a positive association between age, different sites of work, and communication skills with verbal abuse experience. However, interestingly enough, it shows that there is only a slight association between sociodemographic factors such as gender, marital status, and educational level, but as a central factor, all of it is combined with the “own person”, namely their attitudes, behavior, inner positions, self-consciousness, and self-value. Workplace violence among HCPs may be connected to personal values in one’s work. In other words, when a person does something over and over again that he or she does not identify with, believe in, or feel obligated to do, they risk developing an existential vacuum. This contributes to a certain degree of psychological burnout, annoyance, and obstacles in the workplace [36].

It is worth noting that good facial expressions, as well as good communication skills, help to prevent nonverbal abuse and enhance their safety, especially during crises. In physicians, communication skills and age are the only contributing factors to the same type of abuse. Conducting a job rotation approach leads to motivating nurses and enhancing their overall satisfaction and this could ultimately decrease the conflict between nurses and patients, as well as for customers [37].

We acknowledge that these findings are not representative of all HCPs in Jordan during the COVID-19 health emergency. Potential biases such as response bias, the translation of the study tools to the Arabic language, and the sample size are considered barriers to the generalization of the results.

## 5. Conclusions

In general, our results on workplace violence reflect the reality that HCPs have suffered during the COVID-19 pandemic. A high prevalence of workplace violence is noted among HCPs in Jordan compared with before the pandemic, which highlights the importance of promoting public awareness during crises. Since physicians are more likely to experience workplace violence than nurses, this can provide significant cues for setting priorities and strategies in the first place. Moreover, hospital managers should encourage physicians and nurses to formally report any type of violence against them to ensure compliance with the healthcare system. Serious measures should be implemented to deal with workplace violence, especially verbal violence, which has proven to be prevalent. These measures include developing policies and procedures and improving intervention strategies to reduce or even prevent violence. Our quantitative data suggested that communication skills are mainly associated with verbal and nonverbal abuse. To tackle the problem of violence among HCPs during crises, one should pay attention to the psychological aspect of staff who have been subjected to workplace violence by providing comprehensive guidance, training courses to report violence, and supporting them to improve their ability to deal with the stress and tension associated with violence in the workplace.

## Figures and Tables

**Table 1 behavsci-12-00106-t001:** Sample characteristics of nurses (*n* = 190) and physicians (*n* = 102).

Category	Nurses *n* (%)	Physicians *n* (%)	Chi-Square	*p*-Value
General characteristics				
Gender				
Male	71 (37.4)	55 (53.9)	6.78	0.009 **
Female	119 (62.6)	47 (46.1)		
Marital status				
Single	72 (37.9)	77 (75.5)	42.33	0.001 ***
Married	118 (62.1)	25 (24.5)		
Site of work				
Emergency room	51 (26.8)	17 (16.7)		
Adult floor (medical/surgical)	58 (30.5)	49 (48.0)		
Pediatric floor	33 (17.4)	16 (15.7)	16.21	0.003 **
Intensive care unit	37 (19.5)	9 (8.8)		
Psychiatric unit	11 (5.8)	11 (10.8)		
Age				
20–30 years	84 (44.2)	53 (52.0)		
31–40 years	83 (48.9)	44 (43.1)	20.98	0.001 ***
>40 years	13 (6.8)	5 (4.9)		
Educational level				
Diploma degree	21 (11.1)	0 (0)		
Bachelor’s degree	143 (75.3)	41 (40.2)	69.92	0.001 ***
Postgraduate degree	26 (13.7)	61 (59.8)		

Note. ** *p* < 0.01. *** *p* < 0.001.

**Table 2 behavsci-12-00106-t002:** Occurrence of workplace violence, security precautions, and communication skills among nurses (*n* = 190) and physicians (*n* = 102).

Items	Nurses *n* (%)	Physician *n* (%)	*t*-Test	*p*-Value
WORKPLACE VIOLENCE				
1-Have you been subjected to violence in the past year?				
NO	86 (45.3)	31 (30.4)	3.10	0.002 **
once	44 (23.2)	22 (21.6)		
more than once	60 (31.6)	49 (48.0)		
2-Is there anyone in your department besides yourself who has been subjected to violence over the past year?				
Yes	148 (77.9)	83 (81.4)		
No	42 (22.1	19 (18.6)	0.68	0.49
3-How often are you subjected to violence?				
Rarely	104 (54.7)	39 (38.2)		
Once a month	31 (16.3)	24 (23.5)		
More than once a month	44 (23.2)	30 (29.4)	2.40	0.017 *
Almost every shift	11 (5.8)	9 (8.8)		
4-What type of violence have you been subjected to?				
Physical abuse				
Yes	21 (11.1)	20 (19.6)	2.03	0.014 *
No	169 (88.9)	82 (80.4)		
Verbal abuse				
Yes	132 (69.5)	89 (87.3)	3.42	0.001 ***
No	58 (30.5)	13 (12.7)		
Non-verbal abuse				
Yes	119 (62.6)	69 (67.6)	0.91	0.36
No	71 (37.1)	33 (32.4)		
5-If yes, did you ever formally report it?				
Yes	75 (39.5)	27 (26.5)	2.21	0.028 *
No	115 (60.5)	75 (73.5)		
Security precautions				
1-Are the precautions against violence at your department?				
Yes	85 (44.7)	31 (30.4)	2.55	0.001 **
No	105 (55.3)	71 (69.6)		
2-Are there security guards at your department?				
Yes	46 (24.2)	41 (40.2)		
No	144 (75.8)	61 (59.8)	2.74	0.007 **
3-Is there a police checkpoint at your department?				
Yes	6 (3.2)	5 (4.9)		
No	184 (69.8)	97 (95.1)	0.75	0.45
4-Are there metal detectors?				
Yes	8 (4.2)	5 (4.9)		
No	182 (95.8)	97 (95.1)	0.28	0.78
5-Are there security cameras?				
Yes	160 (84.2)	70 (68.6)	2.99	0.003 **
No	30 (15.8)	32 (31.4)		
6-Are there any precautions that prevent entry into the department?				
Yes	65 (34.2)	17 (16.7)	3.41	0.001 ***
No	125 (65.8)	85 (83.3)		
Communication skills				
High	78 (41.1)	61 (61.8)	4.93	0.001 ***
Moderate	52 (27.3)	35 (32.3)		
Low	60 (31.6)	6 (5.9)		
Overall (Mean ± Standard deviation)	3.20 ± 0.99	3.84 ± 0.73		

Note. * *p* < 0.05. ** *p* < 0.01. *** *p* < 0.001.

**Table 3 behavsci-12-00106-t003:** Multinomial logistic regression analysis of violence types among nurses (190) and physicians (102).

Type of Abuse	Nurses Frequency *n* (%) *	Nurses (190) Chi-Square Value	*p*-Value *	Physicians Frequency *n* (%) *	Physician (102) Chi-Square Value	*p*-Value *
Physical abuse General characteristics						
Gender						
Male	11 (52.4)	0.35	0.55	10 (50)	0.01	0.91
Female	10 (47.6)			10 (50)		
Marital status						
Single	5 (23.8)	0.14	0.70	7 (35)	4.02	0.04 *
Married	16 (76.2)			13 (65)		
Site of work						
ER	3 (14.3)	3.89	0.42	3 (15)		
Wards	7 (33.3)			8 (40)		
Pediatric floor	1 (4.8)	17.99	0.001 ***	2 (10)	12.20	0.02 *
ICU	4 (19)			1 (5)		
Psychiatric unit	6 (28.6)			6 (30)		
Age (Years)						
20–30 Y	5 (23.8)			7 (35)		
31–40 Y	15 (71.4)	1.26	0.52	13 (65)	8.91	0.01 **
>40 Y	1 (4.8)			0		
Educational level						0.01 **
Diploma				0		
B.Sc.	1 (4.8)	3.98	0.13	5 (25)	6.38	
Postgraduate	16 (76.2)			15 (75)		
	4 (19)					
Communication skills		46.62	0.21		39.28	0.09
Verbal abuse General characteristics						
Gender						
Male	56 (42.4)	1.07	0.30	48 (53.9)	0.18	0.66
Female	76 (57.6)			41 (46.1)		
Marital status						
Single	51 (38.6)	2.34	0.12	66 (74.2)	0.01	0.98
Married	81 (61.4)			23 (25.8)		
Site of work						
ER	43 (32.6)	19.60	0.001 ***	15 (16.9)	26.14	0.001 ***
Wards	39 (29.5)			41 (46.1)		
Pediatric floor	17 (12.9)			15 (16.9)		
ICU	25 (18.9)			9 (10.1)		
Psychiatric unit	8 (6.1)			9 (10.1)		
Age (Years)						
20–30 Y	55 (41.7)	8.98	0.01 **	42 (47.2)	14.95	0.001 ***
31–40 Y	69 (52.3)			42 (47.2)		
>40 Y	8 (6.1)			5 (5.6)		
Educational level						
Diploma	15 (11.4)	3.47	0.17	0	0.30	
B.Sc.	96 (72.7)			33 (37.1)		
Postgraduate	21 (15.9)			56 (62.9)		
Communication skills	69.29		0.01 **		62.40	0.001 ***
Non-verbal abuse General characteristics						
Gender						
Male	35 (29.4)	5.70	0.01 **	34 (49.3)	0.11	0.74
Female	84 (70.6)			35 (50.7)		
Marital status						
Single	47 (39.5)	0.20	0.65	49 (71)	1.52	0.22
Married	72 (60.5)			20 (29)		
Site of work						
ER	31 (26.1)	3.89	0.42	12 (17.4)	4.97	0.29
Wards	37 (31.1)			32 (46.4)		
Pediatric floor	20 (16.8)			10 (14.5)		
ICU	25 (21)			5 (7.2)		
Psychiatric unit	6 (5)			10 (14.5)		
Age (Years)						
20–30 Y	57 (47.9)	1.14	0.56	37 (53.6)	6.89	0.03 *
31–40 Y	53 (44.5)			27 (39.1)		
>40 Y	9 (7.6)			5 (7.2)		
Educational level						
Diploma	13 (10.9)		0.43	0	1.35	0.25
B.Sc.	92 (77.3)	1.68		26 (37.7)		
Postgraduate	14 (11.8)			43 (62.3)		
Communication skills		68.05	0.01 **		42.92	0.04 *

Note. *: Percentage within profession. ER: Emergency room. Wards: Medical/Surgical. ICU: Intensive care unit. B.Sc.: Bachelor’s degree. Ref: Reference. *p*-value of the Wald Chi-square statistics; * *p* < 0.05. ** *p* < 0.01. *** *p* < 0.001.

## Data Availability

The data presented in this study are available on request from the corresponding author.

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
