# Peer review of "Workplace Violence among Healthcare Providers during the COVID-19 Health Emergency: A Cross-Sectional Study"

_behavsci, 2022, doi:10.3390/bs12040106_

Round 1

Reviewer 1 Report

see attached file

Author Response

Dear editor,

We would like to thank you and both reviewers for their valuable comments and feedbacks. Point-by-point responses to reviewers are listed below.

Reviewer #1: Summary: The authors of the paper conducted a survey in which doctors and nurses of a hospital in Jordan were contacted and asked about their experiences with workplace violence during a three month period in 2019. The authors’ goal is to: a) document the presence of workplace violence and b) identify important correlates with violence (e.g., communication skills, presence of workplace security).

General impressions: Overall, I think that the research question and data efforts warrant publication in the journal. I do have a few comments that I would like the authors to address. Most of my comments request clarification or re-phrasing to allow for more precise statements. A lot of my comments have to do with poor grammar.

Comment 1:

  1. Title:
  2. why are not all words capitalized? some are (Violence) whereas some are not (providers)
  3. Avoid using “the biggest health emergency” since it is probably debatable how that can be

measured but instead just mention “during the COVID-19 Health Emergency” – recommend

changing this throughout the paper

Response 1:

Thank you for your valuable comments. In response to these comments, all words are now capitalized in our title. Furthermore, we changed “the biggest health emergency” to “during the COVID-19 health emergency” as your suggestion.

Comment 2:

  1. Abstract:
  2. line 16 on page 1: barrier to what? entry into profession? please edit to clarify

Response: A new paragraph was added in our revised manuscript as follows “Workplace violence among healthcare providers (HCPs) has a tangible barrier to patient care”.

  1. line 17 on page 1: whose communications skills: healthcare practitioners or those of patients? please edit to make it clearer (this become clearer later on in the paper but should be clear from the start)

Response: A new paragraph was added in our revised manuscript as follows “The purpose of this study was to assess insight into physicians' and nurse’s perceptions of workplace violence and their perceptions of communication skills during the COVID-19 health emergency”.

  1. line 20 on page 1: whose sociodemographic factors? the area where healthcare providers

work? please edit to make it clearer (this become clearer later on in the paper but should be clear from the start)

Response: A new paragraph was added in our revised manuscript as follows “We also sought to assess and compare the association between types of workplace violence, communication skills, and several sociodemographic factors of physicians and nurses including gender, marital status, site of work, age, and educational level during this era”.

  1. line 27 on page 1: the paper lists as one of its conclusions an increase in workplace violence even though the empirical analysis relies on data that span three months during the pandemic but not data prior to pandemic – can you modify this conclusion and simply state that you document large prevalence of workplace violence or/and find evidence of workplace violence is similar setting (Jordan, healthcare providers) before pandemic to justify the claim that violence has gone up?

Response: A new paragraph was added in our revised manuscript as follows “A high prevalence of workplace violence is noted among HCPs in Jordan compared with it before the pandemic, which highlights the importance of promoting public awareness during crises”.

Comment 3:

  1. Introduction (Section 1):
  2. line 35 (page 1): biggest global emergency in 2019

Response: Corrected.

  1. line 37 on page 2: two studies are mentioned that seem to document greater workplace violence for healthcare workers compared to workers in other sectors – can you make clear what it is that your research adds to these two studies? is it a different country? a richer set of controls?

Response: A new paragraph was added in our revised manuscript as follows “However, prior to the coronavirus disease 2019 (COVID-19) pandemic, public sector physicians in both Jordan and China faced different types of workplace violence. The current research results on the prevalence of this phenomenon in the university hospital sector are poorly investigated during this era”.

  1. line 44 on page 1: damaging one’s personality is an odd description; would skip this

Response: Deleted.

  1. line 48 on page 2: document workplace violence with sociodemographic factors – rephrase that as looking at sociodemographic factors that contribute to/correlate with violence?

Response: Corrected as “Sociodemographic factors including gender, marital status, site of work, and age were found to be associated with workplace violence among HCPs”

  1. line 53 on page 2: 2.26 more often/likely? Clarify

Response: corrected as “two times than physicians”

  1. line 54 on page 2: were more likely found to experience…

Response: Corrected.

  1. line 56 on page 2: replace situation with findings

Response: Corrected.

  1. bad grammar: physician was found to be the highest level of workplace violence

Response: Corrected as “where the youngest physician had highest levels of workplace violence

  1. line 58 on page 2: other workplaces - by that you mean still within the healthcare sector or do you mean in other sectors outside the healthcare sector – clarify

Response: Corrected as “The perceived level of workplace violence among HCPs in emergency rooms and psychiatric wards was higher than in other healthcare sectors.”

  1. last sentence in the first paragraph on page 2: how will this study better examine the issues

reviewed in the paragraph? explain briefly

Response: Corrected as “This study will better represent the relevant sociodemographic factors of physicians and nurses by exploring their contribution to workplace violence during the era of COVID-19.”

  1. second paragraph from the top on page 2: introduce the paragraph as describing hypothesis that you are interested in testing with your data (on the role played by communication skills)

Response: Corrected as “We hypothesized that higher perceived level of communication skills among HCPs can provide higher productivity and success, lower anxiety levels, higher patient satisfaction, better decision-making, and lower workplace violence [14]. In contrast, we hypothesized that lower communication skills among HCPs may anticipate higher physical violence, lower motivation and commitment, and may affect the quality of care and services provided to patients [15, 16].

Comment 4:

Section 2:

  1. line 93 (page 2): how were outliers identified? please briefly mention/clarify

Response: Corrected as “A total of 13 surveys were excluded due to incomplete surveys, extreme values, or contradictory answers”.

  1. line 108 in page 3: which adopted by… bad grammar – please fix.

Response: Corrected as “which was adopted from…..

  1. measure of violence: as described in the second paragraph from the top on page 3 the measure of violence relies on raw measures that identify incidence, frequency, and violence type; would it be possible to look at these raw measures as y to get more detailed insight as to what is going on that an aggregated measure captured by a scale might not provide?

Response: Corrected as “Table 2 reveals the detailed items of workplace violence scale”

  1. line 124 on page 3: which was adopted from Axboe…

Response: Corrected as “which was adopted from Axboe…..

Comment 5:

  1. Section 3:
  2. line 144 on page 4: during the year of... just state month and year when the survey was conducted.

Response: Corrected as “We distributed self-reported surveys from September to November 2020”.

  1. line 157 on page 4: were pursuing suggests that they are still in school – just mention that they had x type of education

Response: Corrected as “and 61 had higher educational levels”.

  1. line 170 on page 6: bad grammar/please rephrase - We found a higher proportion of physicians and nurses in violence

Response: Corrected as “Physicians (48%) experienced workplace violence more than nurses (31.6%)”.

  1. line 172 on page 6: verbal abuse was most common… (not exhibited the highest – the highest what?) please re-write for other sentences in the same paragraph

Response: Corrected as “verbal abuse was the most common

  1. line 173 on page 6: you mention significant differences between doctors and nurses but do not elaborate/mention what these differences are – please add details/briefly clarify what you mean by differences

Response: Corrected as “Results indicated that nurses are often physically abused, while physicians are often verbally abused”.

  1. line 177 on page 6: grammar issue - there were no presence

Response: Corrected as “there was no presence”.

  1. sections 3.3.1 and 3.3.2 on pages 7 and 8: describe the nature of correlations (positive vs negative) in words rather than focusing on numbers that tell us only about statistical significance

Response: We deleted and described the nature of correlation as your suggestion.

Response: In 3.3.1. Corrected as “Our results indicated that marital status, site of work, age, and educational level were positively associated with physical abuse. The model also showed that site of work, age, and communication skills were positively associated with verbal abuse. Finally, age and communication skills were positively associated with nonverbal abuse among participating physicians”.

Response: In 3.3.2. Corrected as “We found that site of nurses’ work was positively associated with physical abuse. Moreover, site of work, age, and communication skills were positively associated with verbal abuse. Gender and communication skills were positively associated with nonverbal abuse among participating nurses”.

Comment 6:

  1. Section 4:
  2. line 216 (page 8): rewrite “It is not surmised that HCPs are suffering with violence in their workplace that it becomes challenging and difficult to do their regular duties and tasks, thereby affecting their patient’s life.”

Response: Corrected as “HCPs have experienced more than one type of workplace violence during the COVID-19 health emergency. This dilemma becomes challenging to HCPs and directly affects their regular duties and tasks, thereby affecting their patient’s life.”

  1. line 223 on page 8: bad grammar - nurses generally are less contacted with patients

Response: Corrected as “nurses have less contact with patients than physicians who directly involve with patients”.

  1. line 225 on page 8: rewrite to say that your findings can help to explain evidence in other studies of higher turnover and decreased productivity of Jordan physicians

Response: Corrected as “Our findings can help to explain the evidence in other studies of higher turnover and lower productivity of Jordan physicians.”

  1. line 232 on page 8: bad grammar - This finding is in consistent

Response: Corrected as “This finding is consistent with”

  1. bottom paragraph on page 8: it seems confusing that communication skills which should help doctors diffuse any tensions that might lead to violence do not seem to help in this context – can the authors say anything about potential problems linked to the fact that these skills are self-reported and that this might lead to a bit contradictory finding?

Response: Corrected as “Potential explanations are linked to the fact that these skills are self-reported and that might lead to a bit contradictory finding.”

  1. top paragraph on page 9: grammar is very awkward – recommend that every sentence is

checked and edited to improve grammar

Response: Corrected as “We found that verbal abuse was the most common abuse among both participating samples in this study. This pattern of violence was previously reported among HCPs in Switzerland [30]. Among physicians, different sociodemographic factors were positively associated with physical abuse including marital status, age, educational level, and site of work. They experience this type of violence more frequently than nurses because of their workload, time pressure, critical condition of COVID-19 patients, and an increased number of deaths. While in nurses, only site of work factor was associated with physical abuse and this can be related to workplace factors such as emergency room and psychiatric unit.

  1. lines 260 through 264 on page 9: please re-write since I do not understand what you are

trying to say here

Response: Corrected as “Workplace violence among HCPs may be connected to personal values in one’s work. In other words, when a person does something over and over again that he or she doesn't identify with, believe in, or feel obligated to take care of the sick person, they risk developing an existential vacuum. This contributes to a certain degree of psychological burnout, annoyance, and obstacles in the workplace”.

  1. line 271 on page 9: representative of…

Response: Corrected.

Comment 7:

Section 5:

  1. line 276 (page 9): have suffered

Response: Corrected.

  1. line 277 on page 9: bad grammar - seems to be increased

Response: Corrected.

  1. line 279 on page 9: should encourage without be

Response: Corrected.

  1. line 282 on page 9: prevalent rather than high

Response: Corrected.

  1. line 285 on page 9: drop be in “should be encouraged physicians”

Response: Corrected.

  1. line 288 on page 9: staff rather than cadres

Response: Corrected.

Reviewer 2 Report

In the manuscript, the authors reported a cross-sectional study on workplace violence among HCPs during the COVID-19 pandemic. The topic is important and the findings potentially have implications for healthcare and public health. However, there are several aspects that need to be improved.

  1. As being pointed out in the limitation section, it is unclear is the sample is representative. The recruitment method should be further justified.
  2. Hypotheses were no clearly stated, and the analyses were exploratory. Without hypothesis testing, the findings' validity is questionable. It is suggested to form hypotheses regarding the relationship between communication skills and workplace violence.
  3. It is not clear why physicians and nurses were two separate groups in the study. It is not surprise that physicians would experience more workplace violence than nurses, but what is the implication of this finding?
  4. It is necessary to compare the present findings with some population based data. Otherwise, the ratios of HCPs who have experienced workplace violence is less interpretable.
  5. Although the study is cross-sectional, the present results should be compared with longitudinal data, given the focus on the COVID-19 pandemic in the manuscript. Because the effects of the pandemic can only be investigated through comparisons with non-pandemic periods. 

Author Response

Dear editor,

We would like to thank you and both reviewers for their valuable comments and feedbacks. Point-by-point responses to reviewers are listed below.

Reviewer #2: 

In the manuscript, the authors reported a cross-sectional study on workplace violence among HCPs during the COVID-19 pandemic. The topic is important and the findings potentially have implications for healthcare and public health. However, there are several aspects that need to be improved.

Comment 1:

As being pointed out in the limitation section, it is unclear is the sample is representative. The recruitment method should be further justified.

Response 1:

Thanks for this comment, therefore, we added more details regarding recruitment method as follows “All physicians and nurses in the selected hospital in Amman-Jordan were invited to participate in our self-reported survey. The selected hospital has a sufficient number of physicians and nurses and it is considered a center for COVID-19 patients. Exclusion criteria were physicians and nurses who had maternity leave, did not hold a degree, unpaid leave, and long periods of sick leave. The total number of physicians in the selected hospital was about 350, while the number of nurses was 850 approximately [17, 2]. Sample size was calculated based on the precision of 0.04 with the confidence interval of 95%. In this regards, a total of 114 participants would be representative in each profession [18]. Based on the number of physicians to nurses, we distributed 150 and 250 self-reported surveys, respectively. A convenience sampling procedure was selected to recruit the study sample.”

Comment 2:

Hypotheses were no clearly stated, and the analyses were exploratory. Without hypothesis testing, the findings' validity is questionable. It is suggested to form hypotheses regarding the relationship between communication skills and workplace violence.

Response 2:

As you will see in our revised manuscript, we clearly stated study questions at the end of introduction section as follows:

1.1.      Study questions

  • What is the occurrence of workplace violence and the perceived level of communication skills among physicians and nurses during the COVID-19 health emergency?
  • Is there an association between the types of workplace violence and communication skills among physicians and nurses during the COVID-19 health emergency?
  • What are the sociodemographic factors affecting workplace violence among physicians and nurses during the COVID-19 health emergency?

Comment 3:

It is not clear why physicians and nurses were two separate groups in the study. It is not surprise that physicians would experience more workplace violence than nurses, but what is the implication of this finding?

Response 3:

The first part in your comment was added in our revised manuscript as follows “Recruited samples were selected based on their strenuous job with COVID-19 patients and based on their extensive contact with patients and patients' families during the COVID-19 health emergency. This study will highlight the occurrence of workplace violence among medical professionals (physicians vs. nurses) during the COVID-19 health emergency”.

The implication of findings is represented in our revised manuscript under conclusion section as follows “Since physicians are more likely than nurses to experience workplace violence, it can provide significant clues for setting priorities and strategies in the first place.”

Comment 4:

It is necessary to compare the present findings with some population based data. Otherwise, the ratios of HCPs who have experienced workplace violence is less interpretable.

Response 4:

Thanks. We compare our results with previous studies (non-pandemic periods) findings in our revised manuscript. Please see our result section.

i.e., a new paragraph was added “Comparing our results with non-pandemic period, previous studies in Jordan [26, 27] showed that nurses experienced verbal violence (63.9%) compared to those in our study (69.5%). This study highlights the increase in verbal abuse during the COVID-19 health emergency.”

Comment 5:

Although the study is cross-sectional, the present results should be compared with longitudinal data, given the focus on the COVID-19 pandemic in the manuscript. Because the effects of the pandemic can only be investigated through comparisons with non-pandemic periods.

Response 5:

Thanks for pointing this issue. In response to this comment a new paragraph was added in our revised manuscript as follows “Our finding is in agreement with a previous longitudinal study in Italy [29], which found that physicians suffer a higher occurrence of workplace violence than nurses. Moreover, it found that the rate of physical violence among physicians was (19%) compared to those in our study (19.6%) [29]. Longitudinal studies reflect real-world situations of workplace violence in HCPs as well as during the pandemic period.”

We hope now that our revised manuscript is acceptable for publication.

Round 2

Reviewer 1 Report

na